# The Molecular Composition of Peat Organic Matter and Prospects for Its Use in Agriculture

Evgeny Lodygin [1],*, Roman Vasilevich [1] and Evgeny Abakumov [2]

1  Institute of Biology, Komi Science Center, Ural Branch, Russian Academy of Sciences, Kommunisticheskaya St., 28, 167982 Syktyvkar, Russia; vasilevich.r.s@ib.komisc.ru
2  Department of Applied Ecology, Faculty of Biology, St. Petersburg State University, 16th Liniya V.O., 29, 199178 St. Petersburg, Russia; e_abakumov@mail.ru or e.abakumov@spbu.ru
*  Correspondence: lodigin@ib.komisc.ru

**Abstract:** This paper highlights the molecular composition of native peat and preparations of humic substances (HSs) isolated from permafrost hummock peatlands (Histosols) of the forest tundra zone of the European north-east of Russia. The structural and functional parameters of humic—(HAs) and fulvic acids (FAs) of the peatlands studied are determined by the combined action of cryogenic processes, species composition and the degree of peat decomposition, which reflects the climatic conditions during peat formation in the Holocene. The predominance of the proportion of HAs over FAs in the composition of peat, as well as the low acidity of FAs, makes HS-based preparations highly promising for use as organic fertilizers. The high contents of alkyl and carbohydrate fragments in the structure of the studied HSs allow us to recommend them for use in mineral loamy-textured soils.

**Keywords:** humic acids; fulvic acids; organic fertilizers; NMR spectroscopy; Histosols

## 1. Introduction

Under conditions of decreasing production of mineral fertilizers, a reduction in the number of cattle and a reduction in the amount of organic fertilizers applied to agricultural soils, technologies for obtaining humic preparations (humates)—innovative products from the complex processing of carbon-containing raw materials (coal, leonardite, vermicompost, peat, etc.)—are of great importance. The market volume of organic fertilizers made of humates is forecasted to reach several million US dollars by 2031 compared with 2023, with an unexpectedly high average annual growth rate during the forecast period of 2023–2031 [1]. As a rule, the composition of humic preparations is unstable and largely depends on the sources of raw materials and the techniques used to isolate humic substances (HSs). Humic preparations include potassium and/or sodium humates (up to 90%), particularly potassium and sodium salts of high-molecular humic acids (HAs) and fulvic acids (FAs), nitrogen compounds (2–4%), phosphorus, potassium and other macro- and microelements. The composition of humic preparations corresponds with their wide applications in many branches of agriculture, industry and ecology [2,3].

The active production of humic preparations began at the end of the XX century. Currently, the production of humic preparations is increasing in many countries of the world, including Russia [1]. Humate-based fertilizers have stimulating and adaptogenic properties [4]. They are mainly intended for presowing seed treatment and non-root- and root treatment of plants during the growing season. Humate-based fertilizers have been repeatedly tested and showed a positive effect on yield and product quality, physico-chemical characteristics and soil fertility [5]. The effect of HSs is obvious in the initial period of plant development and in the period of intensive biochemical processes, as well as in stressed plant conditions such as high soil acidity, drought and frost, an excess or lack of nitrogen in the soil or soil salinization [6,7].

The Komi Republic has the necessary raw materials to make humic preparations and products to meet the needs of the region. One of these valuable raw materials is peat. The territory of the republic possesses 4840 peat deposits, with an area of about 2.76 million hectares with peat resources of about 7.6 billion tons (40% moisture). However, the main peat resources (88.4%) are only forecasted, but not prospected yet [8]. This situation requires the organization of relevant research to assess actual peat deposits in the Komi Republic. Based on the existing data on the distribution of peat soils over the territory of the Komi Republic, we can judge the total area of raised peat as taking up 4.5% of its area. Raised peats occupy lowland watersheds, gentle slopes and relief depressions. They are large sphagnum peatlands with a peat layer thickness of 1–1.5 m or more. Normally, the upper layer (40–60 cm) of raised peat deposit includes sphagnum peat that has a low decomposition degree (5–20%). The conversion of this peat into humus preparations is an inefficient procedure due to the low content of HSs in the initial raw material.

The raised peat layer formed below a depth of 40–60 cm from the daylight surface has a mean-to-high decomposition degree of plant material (25–30% or more). In this part of the deposit, peat can vary between raised, transitional and lowland types, with a better quality of raw material. This peat can have a significant effect when processed for the needs of agriculture [9].

Lowland peats (peat deposits) are of the greatest importance as sources of humates and HSs [10]. However, in the territory of the republic, they occupy only 0.5% of the soil cover [8]. Lowland peatlands occur mainly in river valleys or near-terrace depressions of floodplains. Lowland peats are natural biogeochemical barriers against the migration of pollutants between soil and ground waters as follows: watershed → river valleys → rivers → seas and oceans [11]. Their reclamation and subsequent use for peat extraction will inevitably lead to a violation of the natural hydrological regime of rivers and water reservoirs and their large-scale pollution in emergency situations, accompanied by the entry into the environment of significant amounts of pollutants (oil and petroleum products, heavy metals, polycyclic aromatic hydrocarbons, etc.).

The aim of our study was to study the molecular structure of HSs of hummock peatlands in the forest tundra zone and to assess the prospects for their use as fertilizer.

## 2. Materials and Methods

### 2.1. Study Area

The study area is located in the forest tundra zone (the Seida River basin) of the Vorkuta Region, Komi Republic, Russia (Figure 1). The territory of the region is characterized by island permafrost. The climate is moderately continental.

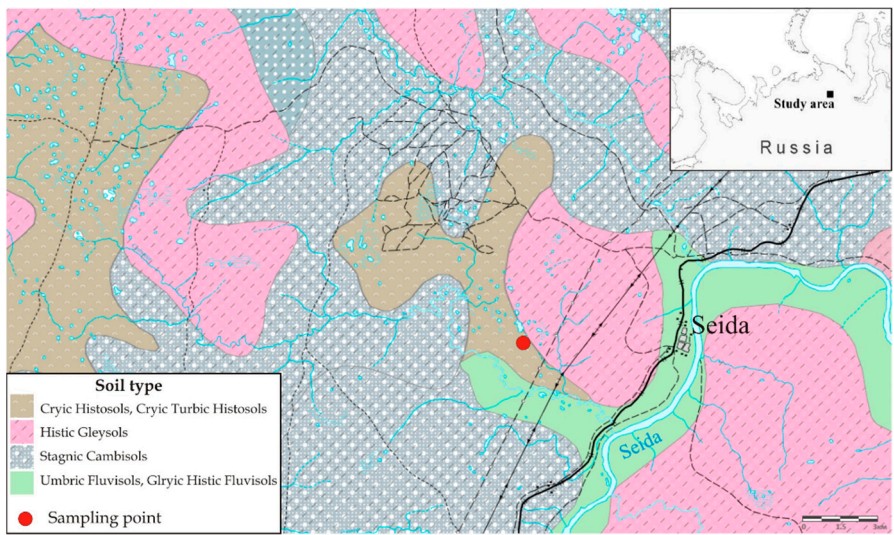

**Figure 1.** Location of sampling area.

The 1960–2019 average mean annual air temperature as observed at the nearby Vorkuta meteorological station is −5.4 °C and the 1960–2019 average annual precipitation is 542 mm [12].

### 2.2. Sampling Sites

The study materials were soils from the hummock–hollow complex: dry-peat permafrost soil of hummocks (Plot-1) and soil of bare peat patches (Plot-2), and Hemic Folic Cryic Histosol and Hemic Folic Cryic Histosol (Turbic), respectively, according to the international soil classification WRB (Figure 2) [13]. Peat samples were collected in triplicate from each layer to a depth of 2.0 m in August 2020. The upper boundary of permafrost in summer is at a depth of 40–60 cm. The peat is dark brown at all depths, with a medium- and high degree of decomposition. The vegetation cover on hummocks includes a complex of wild cloudberry and rosemary–moss–lichen plant communities. The cover on the tops of hummocks is often broken—black peat is uncovered under the influence of wind erosion and frost corrosion, i.e., cryogenic processes. The total area of uncovered peat patches is approximately 8% of the studied hummock–hollow area. The genetic features of the formation of the studied peatlands were previously described in detail by Kaverin et al. [14].

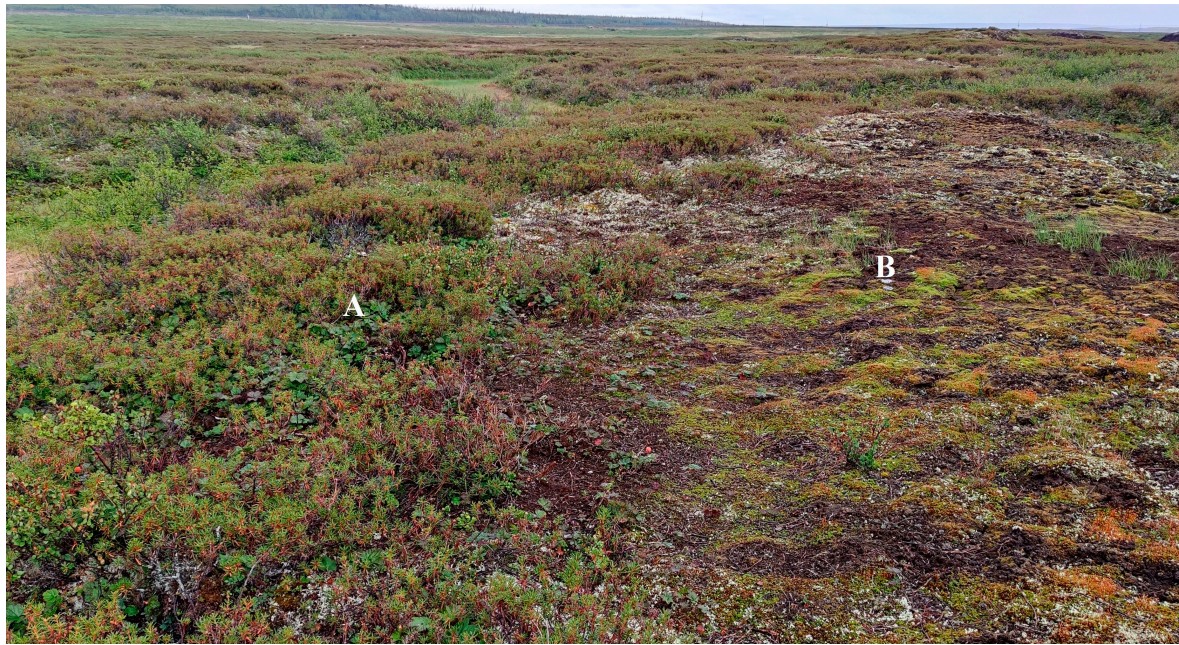

**Figure 2.** The studied peat soils: (**A**) Hemic Folic Cryic Histosol and (**B**) Hemic Folic Cryic Histosol (Turbic).

The physico-chemical characteristics of the soil profiles and the botanical composition of peats are given in Table 1. The temperature regime and morphological description of the studied soil types are given in our previous work [15].

The Hemic Folic Cryic Histosol is located 7 km southwest of the Seida railway station, 67° 03′ N, 62° 56′ E. The moraine plain is low. The soil pit was dug on a hummock with a diameter of 12 m. The vegetation cover is dominated by crowberry, redberry, blueberry, cloudberry, polytrichum and dwarf birch on the edges of the hummock.

The Hemic Folic Cryic Histosol (Turbic) is located 10 m away from the above-described soil. A patch without vegetation, 5 m in diameter, sits on the eastern part of a flat peat hummock (mound). Cryogenic cracks and heaving traces of the upper soil horizon and fragments of dwarf birch bark on the surface can be observed. The patch's edges are covered with lichens and green mosses.

**Table 1.** Properties of peat soils.

| Depth, cm | Peat Layer | Peat Type | Dominant Botanical Species | Degree of Decomposition, % | Ash, % | pH H$_2$O | C/N | Gravimetric Concentrations of HAs, % |
|---|---|---|---|---|---|---|---|---|
| | | | Hemic Folic Cryic Histosol | | | | | |
| 0–10 | Hi | Raised | *Polytrichum* | 20–25 | 5.4 ± 0.3 [1] | 3.75 ± 0.06 | 30.1 ± 2.3 | 16.6 |
| 10–20 | He1 | Raised | Subshrub | 35–40 | 5.9 ± 0.4 | 3.70 ± 0.06 | 24.5 ± 1.8 | 24.8 |
| 20–40 | He2 | Fen | *Carex, Eriophorum* | 30–40 | 4.2 ± 0.3 | 3.74 ± 0.06 | 21.4 ± 1.6 | 21.8 |
| 40–60 | Hef1 | Fen, frozen | *Betula* sp., wood | 30–35 | 6.4 ± 0.4 | 4.34 ± 0.07 | 32.0 ± 2.4 | 26.9 |
| 60–80 | Hef2 | Fen, frozen | *Carex, Menyanthes* | 30–35 | 33.1 ± 1.0 | 4.86 ± 0.07 | 22.4 ± 1.7 | 11.2 |
| 80–100 | Hef3 | Fen, frozen | *Betula* sp., *Carex*, wood | 35–40 | 16.6 ± 0.5 | 4.94 ± 0.07 | 26.8 ± 2.0 | 3.9 |
| 100–150 | Hef4 | Fen, frozen | *Betula* sp., *Carex*, wood | 30–40 | 47.5 ± 1.4 | 4.95 ± 0.07 | 22.2 ± 1.7 | 2.4 |
| 150–175 | Hef5 | Fen, frozen | *Betula* sp., *Carex*, wood | 35–40 | 54.7 ± 1.6 | 5.37 ± 0.08 | 22.0 ± 1.7 | 4.1 |
| 175–200 | Chgf | – | *Betula* sp., *Picea*, grass, wood | >50 | 86.3 ± 2.6 | 5.33 ± 0.08 | 17.2 ± 1.3 | 3.4 |
| | | | Hemic Folic Cryic Histosol (Turbic) | | | | | |
| 0–5 | Hi@ | Raised | *Polytrichum* | 20–25 | 8.0 ± 0.5 | 3.61 ± 0.05 | 28.7 ± 2.2 | 27.6 |
| 5–20 | He1 | Transitional | *Eriophorum*, Subshrub | 40–45 | 4.9 ± 0.3 | 3.42 ± 0.05 | 25.5 ± 1.9 | 24.1 |
| 20–40 | He2 | Fen | *Carex* | 30–35 | 4.6 ± 0.3 | 3.66 ± 0.05 | 26.6 ± 2.0 | 14.2 |
| 40–60 | Hef1 | Fen, frozen | *Carex, Hypnaceous* | 30–50 | 14.1 ± 0.4 | 4.24 ± 0.06 | 23.6 ± 1.8 | 8.5 |
| 60–80 | Hef2 | Fen, frozen | *Carex*, Subshrub | 35–40 | 54.3 ± 1.6 | 5.05 ± 0.08 | 25.0 ± 1.9 | 3.8 |
| 80–100 | Hef3 | Fen, frozen | *Carex* | 35 | 44.9 ± 1.3 | 5.00 ± 0.08 | 24.1 ± 1.8 | 3.4 |
| 100–150 | Hef4 | Fen, frozen | *Betula* sp., *Carex, Picea*, wood | 35–40 | 65.1 ± 2.0 | 5.47 ± 0.08 | 23.4 ± 1.8 | 5.0 |
| 150–175 | Chgf1 | – | *Betula* sp., *Carex, Picea*, wood | 40–45 | 76.9 ± 2.3 | 5.32 ± 0.08 | 21.0 ± 1.6 | 1.9 |
| 175–200 | Chgf2 | – | *Betula* sp., *Carex, Equisetum, Picea*, wood | >50 | 88.2 ± 2.6 | 5.51 ± 0.08 | 18.6 ± 1.4 | 4.3 |

[1] ±S—standard deviation.

The beginning of peat accumulation happened in the middle of the Sub-Atlantic period [15]. The area with spruce and birch growth dominated, with the eutrophic plant communities such as *Carex cespitosa*, *Equisetum*, *Menyanthes*, etc., becoming waterlogged in the absence of permafrost. The quick cooling and decrease in moisture of the middle Sub-Boreal and Early Sub-Atlantic phases in 3754–1357 cal BP (IGRAN 4641, 4640) led to the formation of permafrost [16] and to the dominance of mesotrophic (*Betula nana*, *Eriophorum*) and then oligotrophic communities in the composition of the vegetation cover, such as bryozoan mosses, lichens and dwarf shrubs.

The degree of organic matter humification is more closely related to the soil temperature regime during the warm period of the year. The analysis of the temperature regimes of the studied soils shows that the bare peat patches are subject to very intensive warming of the soil profile in the summer period. The sum of positive temperatures for peat layers from the depth of 10 cm is 2–5 times higher than in the other seasons of the year. In the growing season, biologically active temperatures (>10 °C) reach depths of 20 cm in the soil profile. The soils under bare peat patches have a lower sum of negative temperatures in autumn–winter periods due to the absence of a thermally insulating moss layer, compared with dry peat permafrost soils under vegetated hummocks [15].

Up to the upper permafrost boundary (40–60 cm), peat samples are acidic with a pH value of 3.5–3.9 and ash content of 3.8–6.0%. The chemical composition of the underlying peat layers is characterized by a high proportion of exchangeable calcium and magnesium which account for large increases in pH values up to 5.5 in the lower part of the soil pit. High contents of total Al, Fe and Ca lead to the high ash contents of peat. The fractional composition of humus is due to a significant predominance of HAs over FAs. The mass fraction of HAs reaches a maximum in the upper horizons of the peat (27.6% of the dry mass of the peat sample) and gradually decreases towards the soil-forming rock. Based on the calculation of the Wilcoxon criteria, there are no significant differences in the peat decomposition degree ($p = 0.051$), C/N ratio ($p = 0.95$) and pH ($p = 0.86$) for the two soil pits studied. Thus, soils under bare peat patches are similar to dry permafrost soils under hummocks. They differ only in the temperature regimes of the seasonally thawing layer (STL), which can serve as a simulation model for predicting changes in the structural and functional parameters of HSs due to ongoing climate change [15].

### 2.3. Methods

The HA and FA preparations were isolated from averaged peat samples according to the method widely used by the International Soil Science Society [17]. The hygroscopic moisture content was approximately 7% and the ash content less than 2% in each sample. HSs were extracted from 50 g of dry peat samples using double extractions with 0.1 M NaOH solution in a ratio of 1:10 for complete extraction of HSs. Then, a saturated solution of $Na_2SO_4$ (20% of the extract volume) was added to the resulting extract to coagulate colloidal particles, and the mixture was centrifuged at 10,000 rpm for 1 h in a SIGMA 2–16 KL ultracentrifuge (Sigma Laborzentrifugen GmbH, Osterode am Harz, Germany). HAs were then precipitated with 1 M $H_2SO_4$ to a pH of 2.0. To remove low-molecular-weight compounds, HA preparations were dialyzed and then dried at 35 °C in a laboratory oven with forced convection. FAs were isolated using activated carbon and converted to the $H^+$ form on a KU-2 cation exchange resin.

The $^{13}C$ NMR spectra of air-dried HA and FA preparations and native peat samples were recorded on the NMR spectrometer JNM-ECA 400 (JEOL, Akishima, Japan) with an operating frequency of 100.53 MHz, using the CP/MAS (cross-polarization magic-angle-spinning) solid-phase technique. The sample rotation frequency was 6 kHz, contact time was 5 ms, relaxation time was 5 s and number of accumulations was up to 13,000 scans.

Chemical shifts are presented relatively to tetramethylsilane with a shift of 0 ppm. The adamantane peak at 38.48 ppm was used as the standard. To process the spectra, we used the Fourier transform method with a subsequent correction of the baseline. For quantitative processing of $^{13}C$ NMR spectra, numerical integration over regions corresponding to the

location of C atoms of different molecular fragments and functional groups was performed using the Delta v. 5.0.2 program (JEOL, Akishima, Japan) [18,19].

Statistica v. 6.1 (Dell, Round Rock, TX, USA) software was used for statistical processing of the results.

## 3. Results and Discussion

The studied HSs of tundra peatlands, as well as the native peat samples, have typical absorption spectra (Figure 3). Based on the comparison between the $^{13}$C NMR spectra of HAs and FAs, HA preparations show higher intensity lines corresponding to aliphatic, methoxy and amino groups, and aromatic fragments and lower intensity lines corresponding to carbohydrate fragments.

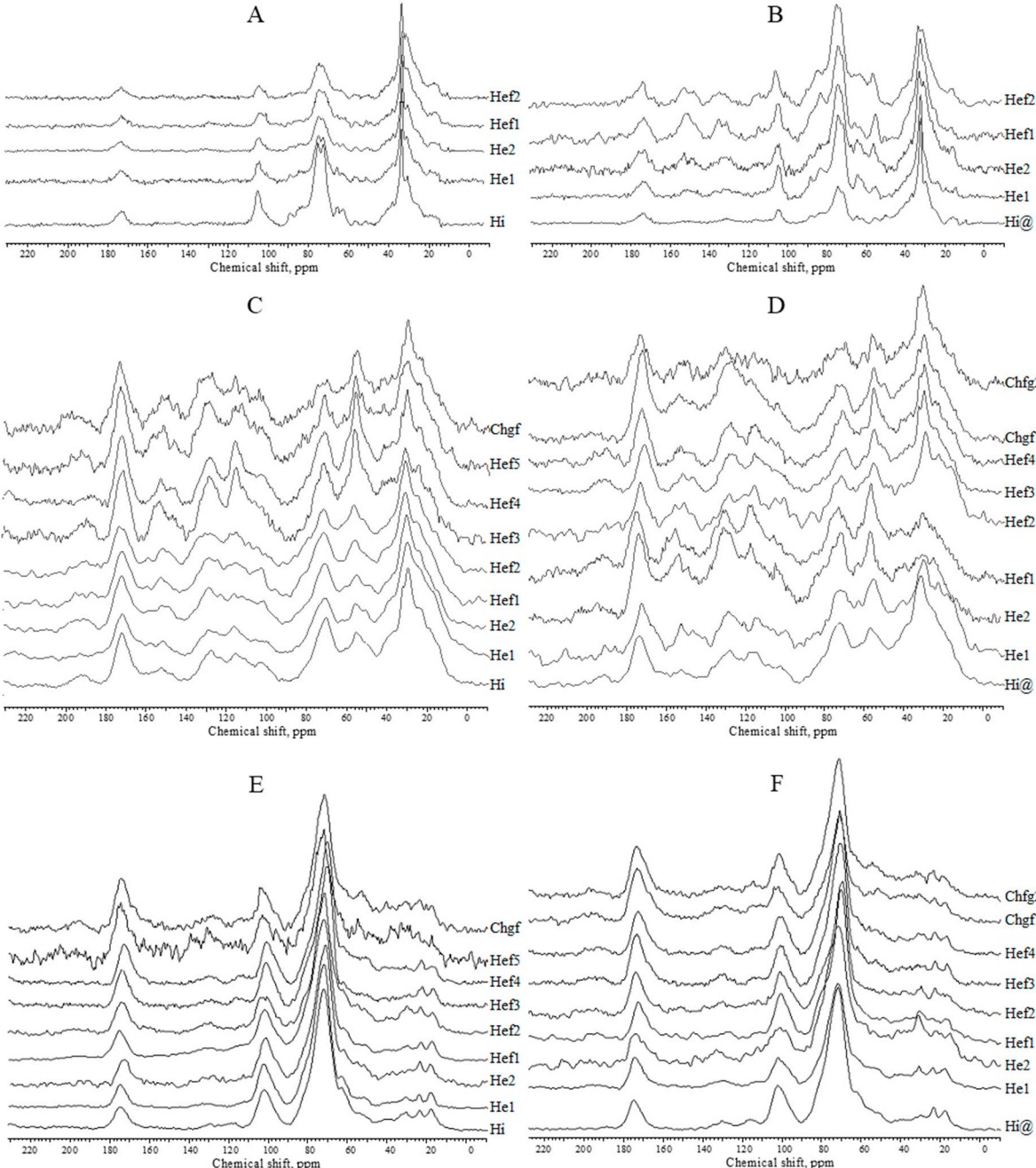

**Figure 3.** CP/MAS $^{13}$C NMR spectra of raw peats (**A**,**B**), HAs (**C**,**D**) and FAs (**E**,**F**) isolated from Hemic Folic Cryic Histosol (**left**) and Hemic Folic Cryic Histosol (Turbic) (**right**).

The analysis of the [13]C NMR spectra of the majority of peat samples and HS preparations demonstrates a high signal intensity in the region of 0–47 ppm corresponding to resonant vibrations of C atoms of unsubstituted aliphatic chains. The high proportion of these atoms in HAs may be associated with the accumulation of plant decay products such as wax resins, phospholipids, glycolipids, cutin, fatty acids, etc. [20]. Methyl groups emitting a signal in the region of 15–18 ppm are confined to highly branched aliphatic structures. Intense lines in the region of 29–33 ppm can be attributed to methylene C atoms of linear chains of paraffins [21], while 35–47 ppm is associated with the presence of C atoms of short aliphatic chains [22]. The minimum signal intensity in the region of 0–47 ppm is observed for HAs from a 40–60 cm layer of Plot-2 soil cut on the permafrost limit, while the maximum is observed for HAs from the upper peat layers (Figure 3 C,D). Down the stratified soil profile, the contribution of branched structures to the general structure of molecules increases, especially for HAs on the STL boundary under bare peat patches (Plot-2).

The main contribution to the resonance of C atoms in the 47–60 ppm region is associated with methoxy groups that dominate the composition of lignin fragments, i.e., syringyl- and guaiacylpropane units [21]. $\alpha$- and $\beta$-carbon atoms from amino acid fragments also resonate in this region of the [13]C NMR spectrum [23].

The intense peak in the region of 60–108 ppm in the [13]C NMR spectra belongs to CH(O) groups of C ring atoms and may also belong to cellulose and other carbohydrate fragments [24]. Moreover, the [13]C NMR spectra of the majority of HAs are identified for signals at 62 ppm, indicating $CH_2O$ groups of hexoses of polysaccharide fragments. This peak is highly intense in the composition of FA preparations. According to [13]C NMR data, the FA structure has an intense signal of 101 ppm of anomeric (acetal) C atoms that also belong to carbohydrate fragments [25].

Alkyl-substituted and/or unsubstituted aromatic C atoms resonate in the region of 108–144 ppm and their analogues replaced by –OH, $-NH_2$ and $-OCH_3$ groups resonate in the region of 144–164 ppm [19]. The signal at 147–149 ppm is typical in spectra of lignin structural blocks and is attributed to O-substituted C atoms of aromatic rings, i.e., syringyl and guaiacylpropane units [21]. The [13]C NMR spectra of HA preparations emit highly intense signals in this region compared with FAs. The range of 164–183 ppm is characterized by its maximum at 172–174 ppm, largely attributed to C carboxyl groups. The maximum can also belong to the carbonyl group of polypeptides [24].

C atoms of carbonyl groups and quinone fragments of aldehydes and ketones contribute very weak signals in the range of 183–190 and 190–204 ppm, respectively.

The content of C atoms of different molecular fragments and functional groups in the composition of native peat and HS is given in Table 2.

**Table 2.** Percentage of C atoms in the main structural fragments of raw peats, HAs and FAs.

| Horizon | Chemical Shift, ppm | | | | | | | |
|---|---|---|---|---|---|---|---|---|
| | 0–47 | 47–60 | 60–108 | 108–144 | 144–164 | 164–183 | 183–190 | 190–204 |
| | Alkyl | O,N-Alkyl | | Aromatic | | Carboxyl, Ester, Amide | Quinone | Carbonyl |
| Hemic Folic Cryic Histosol | | | | | | | | |
| Raw peats | | | | | | | | |
| Hi | 36.8 | 0.9 | 55.5 | 1.7 | 0.7 | 4.2 | 0.0 | 0.1 |
| He1 | 47.6 | 1.1 | 44.4 | 1.2 | 0.7 | 5.0 | 0.1 | 0.0 |
| He2 | 54.1 | 1.5 | 37.4 | 0.9 | 0.2 | 5.4 | 0.2 | 0.2 |
| Hef1 | 57.4 | 1.8 | 34.8 | 1.6 | 0.2 | 3.7 | 0.1 | 0.3 |
| Hef2 | 61.3 | 1.3 | 30.5 | 1.5 | 0.8 | 4.6 | 0.0 | 0.2 |

**Table 2.** *Cont.*

| Horizon | Chemical Shift, ppm | | | | | | | |
|---|---|---|---|---|---|---|---|---|
| | 0–47 | 47–60 | 60–108 | 108–144 | 144–164 | 164–183 | 183–190 | 190–204 |
| | Alkyl | O,N-Alkyl | | Aromatic | | Carboxyl, Ester, Amide | Quinone | Carbonyl |
| *Humic acids* | | | | | | | | |
| Hi | 40.4 | 10.4 | 22.0 | 13.6 | 3.9 | 8.4 | 0.3 | 1.0 |
| He1 | 46.6 | 10.5 | 20.3 | 11.2 | 3.8 | 7.4 | 0.2 | 0.1 |
| He2 | 37.6 | 10.2 | 22.2 | 15.4 | 5.3 | 8.3 | 0.3 | 0.7 |
| Hef1 | 35.5 | 10.4 | 23.2 | 16.9 | 4.9 | 7.7 | 0.7 | 0.7 |
| Hef2 | 35.6 | 11.4 | 22.1 | 16.9 | 4.8 | 7.5 | 0.9 | 0.9 |
| Hef3 | 30.4 | 14.5 | 20.4 | 19.4 | 5.3 | 7.8 | 1.1 | 1.0 |
| Hef4 | 34.8 | 15.8 | 21.5 | 16.0 | 3.1 | 8.0 | 0.5 | 0.2 |
| Hef5 | 32.1 | 11.9 | 19.9 | 20.7 | 5.9 | 8.8 | 0.2 | 0.6 |
| Chgf | 33.2 | 12.2 | 18.5 | 20.6 | 5.1 | 9.1 | 0.1 | 1.4 |
| *Fulvic acids* | | | | | | | | |
| Hi | 12.1 | 6.6 | 74.0 | 1.6 | 0.0 | 5.7 | 0.0 | 0.0 |
| He1 | 7.1 | 4.5 | 80.1 | 2.3 | 0.1 | 5.7 | 0.0 | 0.1 |
| He2 | 11.7 | 7.2 | 72.4 | 3.0 | 0.1 | 5.5 | 0.0 | 0.1 |
| Hef1 | 9.8 | 5.0 | 64.7 | 10.2 | 2.2 | 6.7 | 0.5 | 1.0 |
| Hef2 | 16.2 | 9.0 | 61.2 | 4.5 | 1.0 | 7.3 | 0.1 | 0.6 |
| Hef3 | 10.7 | 5.0 | 70.8 | 3.4 | 0.2 | 9.6 | 0.0 | 0.2 |
| Hef4 | 15.3 | 10.0 | 61.9 | 3.1 | 0.5 | 9.1 | 0.0 | 0.1 |
| Hef5 | 14.5 | 7.7 | 55.0 | 9.2 | 2.2 | 11.1 | 0.0 | 0.2 |
| Chgf | 16.9 | 10.1 | 56.4 | 3.7 | 0.8 | 11.5 | 0.0 | 0.5 |
| *Hemic Folic Cryic Histosol (Turbic)* | | | | | | | | |
| *Raw peats* | | | | | | | | |
| Hi@ | 50.6 | 4.1 | 33.2 | 5.8 | 1.2 | 4.9 | 0.0 | 0.2 |
| He1 | 33.1 | 2.1 | 49.5 | 6.6 | 2.6 | 5.3 | 0.2 | 0.5 |
| He2 | 40.1 | 4.7 | 40.1 | 5.6 | 4.0 | 5.2 | 0.0 | 0.2 |
| Hef1 | 32.9 | 2.5 | 45.2 | 7.0 | 7.1 | 4.7 | 0.3 | 0.3 |
| Hef2 | 30.7 | 5.4 | 48.2 | 5.7 | 4.5 | 5.2 | 0.2 | 0.1 |
| *Humic acids* | | | | | | | | |
| Hi@ | 40.8 | 10.9 | 22.2 | 14.0 | 2.9 | 8.5 | 0.1 | 0.5 |
| He1 | 36.1 | 12.6 | 24.0 | 14.9 | 4.7 | 7.5 | 0.1 | 0.2 |
| He2 | 19.0 | 8.9 | 24.6 | 25.1 | 10.8 | 9.7 | 0.4 | 1.5 |
| Hef1 | 20.7 | 12.6 | 23.5 | 25.1 | 7.5 | 8.7 | 0.9 | 1.1 |
| Hef2 | 39.3 | 12.2 | 22.8 | 14.2 | 5.0 | 6.1 | 0.2 | 0.1 |
| Hef3 | 38.4 | 12.6 | 18.0 | 17.0 | 3.5 | 8.0 | 0.9 | 1.6 |
| Hef4 | 38.0 | 13.4 | 18.8 | 16.1 | 2.7 | 8.3 | 1.3 | 1.5 |
| Chgf1 | 27.7 | 9.4 | 17.2 | 25.3 | 9.2 | 10.8 | 0.0 | 0.4 |
| Chgf2 | 39.8 | 10.2 | 16.7 | 17.3 | 4.8 | 10.0 | 0.3 | 0.9 |
| *Fulvic acids* | | | | | | | | |
| Hi@ | 11.2 | 6.8 | 69.8 | 4.6 | 0.4 | 6.8 | 0.2 | 0.2 |
| He1 | 17.2 | 8.6 | 59.0 | 4.6 | 1.1 | 8.7 | 0.1 | 0.7 |
| He2 | 24.2 | 8.9 | 54.0 | 5.8 | 0.7 | 6.1 | 0.0 | 0.2 |
| Hef1 | 18.2 | 7.9 | 61.1 | 2.5 | 1.3 | 8.1 | 0.0 | 0.9 |
| Hef2 | 18.3 | 8.1 | 55.8 | 6.8 | 1.6 | 9.4 | 0.0 | 0.1 |
| Hef3 | 15.0 | 7.9 | 57.9 | 3.4 | 1.2 | 12.4 | 0.5 | 1.8 |
| Hef4 | 15.4 | 8.8 | 59.2 | 3.0 | 1.3 | 10.6 | 0.5 | 1.3 |
| Chgf1 | 17.9 | 9.8 | 47.4 | 8.4 | 1.4 | 14.3 | 0.0 | 0.8 |
| Chgf2 | 14.9 | 9.4 | 59.3 | 3.6 | 0.4 | 12.1 | 0.0 | 0.3 |

The [13]C NMR analysis of raw peat samples from both soils demonstrates a significant contribution of organic substances of an aliphatic nature. The proportion of paraffin structures reaches 61% and the proportion of carbohydrate fragments reaches 56%. The composition of peat-forming shrubs and bryophytes is dominated by these structures [26]. In spite of the uniformity of the botanical composition, the distribution of molecular groups and fragments for the two peatlands has significant differences (Table 2). The soil composition under bare peat patches (Plot-2) has a comparatively larger proportion of oxidized "lignin" aromatic structures and methoxy groups.

Based on the analysis of the relative content of structural fragments in HS preparations, carbon atoms of aromatic components make up a low proportion, ranging from 15.2 to 36.3% for HAs and from 1.6 to 12.8% for FAs. In addition, the distribution patterns of this indicator for the two soil profiles exhibit significant differences. The content assessment of functional groups and molecular fragments in HAs of dry-peat permafrost soils indicates a difference in their molecular composition on the transition from upper to lower peat layers, accompanied by an increase in the proportion of aromatic fragments and methoxy groups and a decrease in the relative proportion of alkyl groups (Table 2). This is a direct effect of the relationship of the HA's structural and functional parameters, with the climatic parameters in the Holocene indicating warmer periods of HA formation [15]. There is a trend of a decrease in the proportion of oxidized fragments in the composition of HA aromatic structures.

Within the STL of the soil under uncovered peat patches, the relative proportion of aromatic HA fragments significantly increases with depth. The higher aromaticity and lower proportion of non-oxidized aliphatic fragments of HAs at the STL boundary are probably determined by the dynamic processes of freezing and thawing at the permafrost boundary of soils under bare peat patches. HS solutions' freezing causes HS redistribution caused by the degree of complexity associated with the permafrost destruction of the initial molecules due to the wedging effect of freezing water films [27]. The transformation of HA macromolecules and removal of highly labile fragments from them causes an increase in the proportion of FAs and the aromaticity degree of the initial HAs.

As soon as the soils of bare peat patches (Plot-2) are easily warmed during the meteorological summer, they exhibit a comparatively higher degree of peat decomposition, which increases from 20 to 50% from the upper horizons to the STL boundary. The proportion of sedges in the composition of the peat-forming agents increases to 55%. For soils formed under vegetation, the degree of peat decomposition does not increase as much: from 20–25 to 30–35%. These indicators highlight a deep organic material humification in the acrotelm of soils under bare peat patches that probably started since the permafrost formation in the territory 2000–2500 years ago [16]. There are significant differences in the structural and functional parameters of HAs from STL between the two soils studied, which are directly related to the formation of bare patches on the soil surface. This formation can be either a modern process or, probably, one that occurred several hundred years ago. Humification of sedge peat types is a quick process, which begins already in peat-forming plants and is diagnosed by measuring the amplification of signals from aromatic C atoms [28–30]. Consequently, the STL of permafrost peatlands belongs to a modern stage of organic material transformation that is best manifested in soil pits without vegetation cover (Plot-2). High indicators of the sum of positive and biologically active soil temperatures of bare peat patches actually simulate modern global warming and lead to significant changes in the structural and functional parameters of soil organic matter.

The low differentiation of the indicators of the HS composition below the STL boundary is related to the fact that the intensive peat accumulation during the Early- and Middle Holocene finally led to peat horizons (catothelm) being overlaid with new layers (acrotelm). The further decomposition of organic material slowed down considerably, while the peat decomposition rate remained almost the same. Thus, the lower layers turned out to be a kind of buried soil with slowed-down soil processes, which have been completely stopped since the formation of permafrost in the Sub-Boreal period. For the studied relict 150–175 cm



thick peat layer, there are intense maxima indicating the content of aromatic fragments, both in HAs and FAs (Table 2), dating back to the Holocene climatic optimum [31] about 7000 cal BP (IGRAN 4645, 4646) and reflecting the warmest climatic period of its formation.

The composition of labile components in the HS structure is associated with the reduction regimes and low biological activity of peatlands. Organic matter, which is in a preserved state, transforms at an extremely slow rate. The proportion of labile carbohydrate fragments in the HA composition has a weak tendency to decrease with depth. Similar results were obtained for HS from Latvian raised bogs [26]. Bacteria are thought to be more actively involved in the process of humification of organic matter in peats than in molds, yeasts and actinomycetes. However, for bacteria involved in the decomposition of plant residues, the carbohydrate complex is virtually unavailable.

Peat FAs are enriched in O-containing fragments, and less so in alkyl structures (up to 24.2%). The content of carboxyl groups in HAs and FAs in peatlands exhibits similar values, which is not typical for mineral soils where the mass fraction of –COOH in FAs significantly exceeds that of HAs [18]. In addition, the pK values of carboxyl groups of peat FAs are higher than those of FAs from Retisols [32,33], giving peat HSs preference for use as organic fertilizers. During the process of FA carboxylation, the proportion of –COOH groups grows down the soil profile and noticeably exceeds their proportion in the composition of HAs from the lower-lying layers. The high proportion of carbohydrate fragments (up to 80.1%) makes FAs a favorable nutrition source for microorganisms compared with HA molecules (up to 24.6%). More than half of the HA molecules consist of hydrophobic thermobiotic aliphatic chains and aromatic fragments. Another distinguishing feature between FAs and HAs is the difference in the content of methoxy groups and amino acid fragments (range of 47–60 ppm), as well as the content of carbonyl groups. The change in the molecular composition of FAs, being a highly labile pool of HSs, is extremely susceptible to soil formation conditions. The differences in the content of alkyl, amino and methoxy groups and carbohydrate fragments of FAs in various peat layers could be due to differences in temperature regimes and the composition of palynological and botanical material during the Holocene. These factors also explain the maximum content of aromatic fragments in FAs from the Hef1 horizon located on the permafrost limit and experiencing dynamic thawing–freezing processes which finally lead to the transformation of HA structures into FA molecules.

The sample of HA and FA preparations was representative, enabling statistical analysis of the molecular structure of the HSs. The analysis included important characteristics of native peat samples, such as botanical composition and degree of peat decomposition. This helped establish the main transformations that HS macromolecules undergo. The correlation analysis was carried out using the Pearson criterion. The importance of factors like aromaticity and the amount of non-oxidized aliphatic fragments was examined by assessing the structural and functional characteristics of HSs in stratified peat horizons. This was carried out using single-factor analysis of variance (ANOVA) and the Fisher criterion to establish correlations between these factors. All statistical calculations were performed using a significance level of $p \leq 0.05$. For the subsequent analysis, particular plant species were divided into peat-forming groups based on the botanical composition data. The content of lignin in the dry ash-free matter of lichens and mosses (8–10%) is less than that in grass (15–20%), shrub and wood (20–30%) vegetation [34]. Based on these data and considering the botanical composition of the peat, we figured out how much lignin was present in the original plant remains.

Methoxy groups are typically found in the phenylpropane fragments of the "lignin" aromatic components. This fact is confirmed by a reliable correlation between the content of methoxy groups and the proportion of oxidized aromatic fragments of HAs and FAs ($n = 36$, $r = 0.50$, $r_{cr} = 0.33$). Aromaticity and the proportion of HA carboxyl groups ($n = 18$, $r = 0.64$, $r_{cr} = 0.47$) also correlate, indicating the main directions of HS evolution due to humification as carboxylation and selection of thermo-biodynamically stable structures.

The significant negative correlation between the content of the non-oxidized aromatic structures and non-oxidized aliphatic fragments of HAs ($n = 18$, $r = -0.95$, $r_{cr} = 0.47$), as well as the dispersion analysis data ($F = 15.77$, $p = 0.0014$) for these factors, cannot be explained only by the processes of "removal" of alkyl structures as a result of "natural selection", but seem to be a consequence of the cyclization of paraffin chains with subsequent dehydrogenation into aromatic rings.

A reliable correlation exists between the degree of peat decomposition and the content of methoxy groups ($r = 0.47$) and carboxyl groups of FAs ($r = 0.61$, $n = 18$, $r_{cr} = 0.47$). A negative reliable correlation was found between the degree of peat decomposition and the proportion of HA carbohydrate fragments ($r = -0.56$). The calculated lignin content is correlated with the oxidized (ligninic) aromatic fraction ($r = 0.49$) and the FA carboxyl group content ($r = 0.52$). It was found that the aromaticity of the HA molecules does not necessarily indicate the degree of degradation of the peat, but rather, depends on its botanical composition.

## 4. Conclusions

Peat HAs and FAs are represented by weakly condensed molecular structures with a low proportion of C atoms from aromatic fragments. The authors have noted the similar content of carboxyl groups in HAs and FAs of peats which distinguishes the organic matter of peatlands from that of mineral soils, where the mass fraction of –COOH groups in the FA structure is significantly higher due to the normally anaerobic conditions in peat soils.

Analysis of the data on the content of molecular fragments and functional groups in the structure of HSs indicates the role of specific soil processes in leading to the transformation of their molecular composition. An STL is found, with a trend of decreasing in relation to the proportion of alkyl groups and increasing in relation to the proportion of aromatic fragments in HAs and carboxyl groups in FAs moving down through the peat soil profile, as a result of their natural evolution. The actively developing processes of selection or transformation of linear aliphatic fragments that are unsubstituted by O,N-atoms into sterically branched structures can provoke the cyclization of paraffin chains into stable aromatic structures. As soon as peatlands are low, biologically active soils from upper to lower horizons and labile carbohydrate fragments are subject to a low degree of transformation. There is a noticeable increase in the proportion of aromatic fragments in the HAs on the STL boundary which may be caused by a persistent influence of low temperatures leading to the removal of aliphatic structures from the macromolecules.

Climatic features during the period of peat layer formation determined the qualitative and quantitative composition of the main peat-forming plants and the degree of degradation of natural biopolymers. According to the analysis of the molecular structure of HSs from peats, the domination of HAs over FAs in peat composition, as well as their low acidity compared with HSs from mineral soils of the cryolithozone, makes the preparations based on them extremely promising for use as organic fertilizers. The high content of paraffin and carbohydrate fragments in the structure of HSs of peats allows us to recommend them for fertilization of mineral loamy-textured soils. This will help avoid the rapid mineralization of the used humic preparations.

In the future, when developing peat resources, special attention should be paid to an integrated approach to the processing of peat material of different compositions and to the use of innovative technologies that cause minimal damage to the environment. Entrepreneurs interested in developing peat deposits should be aware that any, even minor, external impact on peatlands will have a negative impact on the ecological stability of the region's hydrological network. In addition, the use of peat resources for industrial purposes should be very carefully organized and include sustainable nature management techniques to reduce possible negative environmental impacts.

Taking into account the limited knowledge on peat resources in the Republic of Komi, it is necessary to make a preliminary assessment of the qualitative and quantitative composition of peat as a source of humate and humic fertilizers; to estimate the permissible,

environmentally friendly volumes of peat extraction and processing; to identify soil massifs as a source of peat that are suitable from the point of view of ecology and industrial production and to carry out research on the isolation of HA from different types of peat with, keeping in mind HA's properties and the possibility of its use in agricultural production.

**Author Contributions:** E.L. and R.V.: conceptualization, E.A.: funding, R.V.: expedition with field-work and soil sampling, E.L. and R.V. wrote the paper, E.L. and R.V.: analysis of HAs. All authors have read and agreed to the published version of the manuscript.

**Funding:** The work of Abakumov E. was supported by Saint Petersburg State University, project ID: 101662710 (CZ_MDF-2023-1).

**Data Availability Statement:** Not applicable.

**Acknowledgments:** The reported study by Lodygin E. and Vasilevich R. was carried out within the framework of the budget theme of the Institute of Biology (No. 122040600023-8).

**Conflicts of Interest:** The authors declare no conflict of interest.

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
