# Peer review of "The Molecular Composition of Peat Organic Matter and Prospects for Its Use in Agriculture"

_agronomy, doi:10.3390/agronomy13092414_

Round 1
Reviewer 1 Report
This paper characterized the molecular structure of native peat and HS preparations isolated from peat bogs of the tundra area in the European north-eastern part of Russia. It provide some interesting data on humification processes taking place in harsh climatic conditions. The English language of the manuscript requires improvement. The paper is worthy publishing after moderate revision.
Line 23-27: please, reedit this sentence
Line 83: replace “weather” with “meteorological”
Line 83-84: delete the sentence “The average daily sum of positive …”
Line 89: replace “every” with “each”
Line 133: do you mean “above”?
Line 75: replace “EPR Data” with “NMR spectra”
Line 221: I suggest separating table 2 into two tables with separate headers
Line 343-345: this sentence is a summary, but not the conclusion. It should be deleted.
Line 374-382: this conclusion does not follow from the presented research
English language of the manuscript requires improvement
Author Response
Dear reviewer, thank you very much for your detailed work with the paper and for your suggestions and recommendations for improving the text of the manuscript.
We have marked all changes in the text in yellow.
Detailed comments on all suggestions are given below:
Lines 23-27: the sentence has been corrected.
Line 83: the word "weather" has been changed to "meteorological".
Line 83-84: The sentence "The average daily sum of positive..." has been deleted.
Line 89: The sentence has been corrected.
Line 133: Yes, "above" was meant. The sentence has been corrected.
Line 75: The subheading has been deleted.
Line 221: In our opinion, splitting Table 2 is not reasonable as the combined table allows the reader to see and compare the entire data set at once.
Line 343-345: The sentence has been deleted.
Line 374-382: Conclusions have been corrected.
The English has been further checked and improved.
Yours sincerely,
Authors
Reviewer 2 Report
The present article entitled “Molecular Composition of Peat Organic Matter and Prospects for Its Use in Agriculture” have been compiled to study the molecular structure of humic sub stances of hummock peat lands in the forest tundra zone and to assess the prospects for their use as fertilizer. The authors have considered very few parameters to discuss the molecular composition. But compiled in a good way.
However the article needs moderate revision in English . few points are mentioned here
Table-1 if sampling were carried out in replicate then add std. errors or std. deviation
LINE 89-91 “oil sampling from every layer was carried 89 out down to a depth of 2.0 m. The upper permafrost limit in summer is at a depth of 40– 60 cm. Peat throughout the whole profile is dark-cinnamonic, of medium-to-high decom position degree and black mould humus type” Reframe the sentence for clarity
Line -124 check the sentence -- depends on the temperature soil regimes in the warm year season.
Line- 150- it should be “For any study of the sample”,
The present article entitled “Molecular Composition of Peat Organic Matter and Prospects for Its Use in Agriculture” have been compiled to study the molecular structure of humic sub stances of hummock peat lands in the forest tundra zone and to assess the prospects for their use as fertilizer. The authors have considered very few parameters to discuss the molecular composition. But compiled in a good way.
However the article needs moderate revision in English . few points are mentioned here
Table-1 if sampling were carried out in replicate then add std. errors or std. deviation
LINE 89-91 “oil sampling from every layer was carried 89 out down to a depth of 2.0 m. The upper permafrost limit in summer is at a depth of 40– 60 cm. Peat throughout the whole profile is dark-cinnamonic, of medium-to-high decom position degree and black mould humus type” Reframe the sentence for clarity
Line -124 check the sentence -- depends on the temperature soil regimes in the warm year season.
Line- 150- it should be “For any study of the sample”,
Author Response
Dear reviewer, we would like to thank you for your detailed work on the paper and for your suggestions and recommendations for improving the text of the manuscript.
We have marked all changes in the text in yellow.
Detailed comments on all suggestions are given below.
Standard deviation values have been added to Table 1. Peat was sampled in triplicate. An averaged sample was used for the isolation of HA and FA preparations.
Line 89-91: The suggestion has been adapted.
Line 124: The sentence has been corrected
Line 150: The sentence has been corrected.
The English has been further checked and improved.
Yours sincerely,
Authors
Reviewer 3 Report
The research topic is very interesting. Due to changing climatic conditions, the production and use of mineral fertilisers should be greatly reduced or even stopped altogether. However, in spite of the topical relevance of the research undertaken, the authors have not made a solid effort to describe it. Comments on the manuscript follow:
1. Page 2 line 83-84 - surely this is the average daily sum of positive temperatures? Is it the air temperature? The values given are very high.
2. From what time interval (months, years) were the temperature and precipitation values taken?
3. Were samples taken only from the locations marked A and B on the map? Was it one sample or several and the result was averaged - there is no mention of this in the methodology.
4. The authors state that the morphological description and temperature regimes, as well as the detailed species composition of peat-forming agents for the investigated soil types are given in a previous paper. This work should also include such a description.
5 The methodology does not state how many years the study is.
6. The paper lacks a list of abbreviations used.
Author Response
Dear reviewer, thank you very much for your detailed work on the paper and for your suggestions and recommendations for improving the text of the manuscript.
We have marked all changes in the text in yellow.
Detailed comments on all suggestions are given below.
1. Page 2, lines 83-84: The sentence has been deleted.
2. The time interval for which the long-term averages of temperature and precipitation were taken has been added.
3. Peat samples were taken in triplicate. An averaged sample was used for the isolation of HC and FC preparations. Information has been added to sections 2.2. and 2.3.
4. Data on the dominant botanical species for the peat layers have been added to Table 1.
5. Information on the date of peat sampling was added in section 2.2.
6. The abbreviations used have been checked and corrected.
The English language has been further checked and improved.
With kind regards,
Authors